# *Stieleria sedimenti* sp. nov., a Novel Member of the Family *Pirellulaceae* with Antimicrobial Activity Isolated in Portugal from Brackish Sediments

**DOI:** 10.3390/microorganisms10112151

**Published:** 2022-10-30

**Authors:** Inês Rosado Vitorino, Dominika Klimek, Magdalena Calusinska, Alexandre Lobo-da-Cunha, Vítor Vasconcelos, Olga Maria Lage

**Affiliations:** 1Departamento de Biologia, Faculdade de Ciências, Universidade do Porto, Rua do Campo Alegre s/n, 4169-007 Porto, Portugal; 2CIIMAR/CIMAR, Centro Interdisciplinar de Investigação Marinha e Ambiental, Universidade do Porto, Terminal de Cruzeiros do Porto de Leixões, Avenida General Norton de Matos, S/N, 4450-208 Matosinhos, Portugal; 3The Environmental Research and Innovation (ERIN), Luxembourg Institute of Science and Technology (LIST), 41 rue du Brill, L-4422 Belvaux, Luxemburg; 4The Faculty of Science, Technology and Medicine (FSTM), University of Luxembourg, 2 Avenue de l’Université, L-4365 Esch-sur-Alzette, Luxembourg; 5Laboratório de Biologia Celular, Instituto de Ciências Biomédicas Abel Salazar, ICBAS, Universidade do Porto, Rua de Jorge Viterbo Ferreira, 228, 4050-313 Porto, Portugal

**Keywords:** Alcochete, biosynthetic gene clusters, carbohydrate-active enzymes, iChip, novel taxa, organic extraction, bioactivity screenings, *Planctomycetota*, transmission electron microscopy

## Abstract

The phylum *Planctomycetota* is known for having uncommon biological features. Recently, biotechnological applications of its members have started to be explored, namely in the genus *Stieleria*. Here, we formally describe a novel *Stieleria* isolate designated as strain ICT_E10.1^T^, obtained from sediments collected in the Tagus estuary (Portugal). Strain ICT_E10.1^T^ is pink-pigmented, spherical to ovoid in shape, and 1.7 µm ± 0.3 x 1.4 µm ± 0.3 in size. Cells cluster strongly in aggregates or small chains, divide by budding, and have prominent fimbriae. Strain ICT_E10.1^T^ is heterotrophic and aerobic. Growth occurs from 20 to 30 °C, from 0.5 to 3% (*w/v*) NaCl, and from pH 6.5 to 11.0. The analysis of the 16S rRNA gene sequence placed strain ICT_E10.1^T^ into the genus *Stieleria* with *Stieleria neptunia* Enr13^T^ as the closest validly described relative. The genome size is 9,813,311 bp and the DNA G+C content is 58.8 mol%. Morphological, physiological, and genomic analyses support the separation of this strain into a novel species, for which we propose the name *Stieleria sedimenti* represented by strain ICT_E10.1^T^ as the type of strain (=CECT 30514^T^= DSM 113784^T^). Furthermore, this isolate showed biotechnological potential by displaying relevant biosynthetic gene clusters and potent activity against *Staphylococcus aureus*.

## 1. Introduction

The continuous rise of antimicrobial-resistant microbes is currently one of the major global health problems to affect our society [1,2]. With progressively more and more antibiotics losing their effect, we are in need of novel molecules that could have distinct proprieties and mechanisms of action and this way evade the existing antibiotic resistance mechanisms. One appealing strategy to find chemical novelty in nature could be exploring organisms that are still under-researched in the biotechnological field when compared to other well-studied groups such as filamentous fungi and bacteria from the phylum *Actinomycetota* [3]. 

One captivating group to study is the phylum *Planctomycetota*, which together with the phyla *Chlamydiota*, *Verrucomicrobiota* (PVC), and other recently described groups form the super-phylum PVC [4,5,6]. The bacteria belonging to the phylum *Planctomycetota* [7], commonly designated as planctomycetes, are one of the most enigmatic groups in procaryotes [8,9]. This is due to many of their distinct features, such as cell divisions by binary fission and budding through still unknown mechanisms (without FtsZ machinery), complex cellular morphologies, and unique biology [4,8,10,11,12,13,14,15,16]. Additionally, the currently described diversity still does not cover many lineages putatively detected in environmental studies. In the last years, an increase in the number of isolation studies occurred which allowed the cultivation of many new strains and the description of many novel species [10,17,18,19,20,21,22,23,24,25,26,27,28,29,30,31,32,33,34,35,36,37,38,39,40]. Nevertheless, the majority of 16S rRNA gene sequences putatively assigned to the phylum *Planctomycetota* are still associated with uncultured bacteria (data from the SILVA SSU database of full-length 16S rRNA gene sequences (release 138.1 from 27 August 2020) [41]), which reinforces the need for more isolation studies to help to continue to unveil the diversity and biology of this interesting bacterial group.

Taxonomically, the phylum *Planctomycetota* is subdivided into the classes *Phycisphaerae* and *Planctomycetia* [42,43]. Additionally, only with the *Candidatus* status due to the lack of axenically cultured representatives, the anaerobic ammonia-oxidizing (anammox) planctomycetes from the class “*Candidatus* Brocadiia” [44,45,46]. At the present time, the majority of cultured strains and taxonomically described species belong to the class *Planctomycetia* (slightly above 100), while the class *Phycisphaerae* only has 10 species [47]. The current class *Planctomycetia* is subdivided into the orders *Pirellulales* (containing families *Pirellulaceae*, *Lacipirellulaceae,* and *Thermoguttaceae*), *Gemmatales* (family *Gemmataceae*), *Planctomycetales* (family *Planctomycetaceae*) and *Isosphaerales* (family *Isosphaeraceae*) [27,47]. In its turn, the family *Pirellulaceae* is the family with the current higher number of cultured representatives and species with validly published names [47]. Members of the class *Planctomycetia* divide by budding and are, in its majority, mesophilic, aerobic, neutrophilic, rosette-forming, white/beige to pink pigmented and capable of degrading diverse carbon and nitrogen sources [27,47]. Ecologically, members of orders *Pirellulales* and *Planctomycetales* are mainly distributed in marine habitats while planctomycetes from the orders *Isosphaerales* and *Gemmatales* are almost exclusively associated with freshwater environments [47]. 

In the last years, *Planctomycetota* have started to emerge as possible reservoirs for novel natural products. They live in competitive environments, have complex lifestyles, big genome sizes, high number of putatively encoded giant proteins and large coding regions in the genome with still unknown functions, thus making them good candidates for the production of novel classes of secondary metabolites [8,47,48]. Furthermore, diverse genetic and bioactivity screenings (antimicrobial and anti-cancer) have highlighted the biotechnological potential of these bacteria [10,26,49,50,51,52,53,54,55,56]. As recently reviewed [48], studying less-explored groups such as the planctomycetes can be a good strategy to find chemically distinct molecules with possible health-boosting abilities. At this point, the number of described planctomycetes-derived compounds is still scarce, only consisting of three types of secondary metabolites [48]: carotenoids [57], an aryl halide 3,5-dibromo-*p*-anisic-acid molecule with algicide effect [58] and novel *N*-acyl amino acids compounds designated as stieleriacines [52,56]. With a particular interest in the biotechnological field are the stieleriacines, which showed mild antimicrobial effects on Gram-positive bacteria and were hypothesized as environmentally relevant by altering marine biofilm composition [52,56]. These six novel *N*-acylated tyrosines were isolated from two of the three current valid species of the recently described genus *Stieleria* (from the family *Pirellulaceae*), *S. maiorica* and *S. neptunia* [52,56] (but not from *S. varia* [59]). The limited number of characterized planctomycete metabolites is mainly due to the still low number of strains available in axenic cultures in the laboratory, in comparison with other well-studied bacterial groups such as the *Actinomycetota*, *Pseudomonadota, Bacillota,* and *Bacteroidota*. Planctomycetes are also slow-growing bacteria with relatively low biomass production that often need complex culture media to grow, which also limits the production of secondary metabolites in large amounts required for their isolation and characterization. Increasing the number of cultured planctomycete strains and optimizing their culturing and metabolite extraction protocols is thus essential to improve the knowledge available in the planctomycete biotechnological research field. 

Recently, we applied a novel culturing technique on samples collected in two regions of Portugal (north coast and Tagus river estuary) that allowed the isolation of planctomycetes from different taxa, including novel species [26,29,36]. In this polyphasic study, we formally describe a novel isolate from the genus *Stieleria* [52,56] previously isolated from the Tagus estuary, strain ICT_E10.1^T^. Based on genomic, morphological, and physiological analyses, we propose a novel species, for which we suggest the name *Stieleria sedimenti* represented by strain ICT_E10.1^T^ (=CECT 30514^T^= DSM 113784^T^) as the type strain. Furthermore, we explored the biotechnological capability of strain ICT_E10.1^T^ by genome mining and antimicrobial screenings, which revealed the presence of relevant biosynthetic gene clusters and potent antimicrobial activity against *Staphylococcus aureus*. This study adds knowledge on the existing planctomycete diversity and reinforces the biotechnological potential previously attributed to the genus *Stieleria* [56]. It also suggests that our isolate, strain ICT_E10.1^T^, is a promising organism regarding the production of possibly novel metabolites. 

## 2. Materials and Methods

### 2.1. Isolation

Strain ICT_E10.1^T^ was isolated from sediments that were collected in the Tagus river estuary, in Alcochete (38°45′24.9″ N 8°57′58.9″ W), in the framework of a previous planctomycete isolation study [26]. The sampling occurred on the 13th of May of 2021. The methodology used for the isolation of this strain was based on the iChip technique and performed as described previously [26,29]. Briefly, the environmental inoculum was obtained by mixing 20 g of the collected sediment in 10mL of sterile water and then diluted 1:10 in agarized sea water (0.3% *w/v* agar) supplemented with antibiotics and an anti-fungal (200 mg/L ampicillin, 1000 mg/L streptomycin and 20 mg/L cycloheximide) to help select planctomycete growth and prevent fungal contaminations. Two hundred µL of this mixture were poured into each well of a 96-Well Filtration Plate MultiScreen^®^ (Millipore, Burlington, MA, USA) which contains a filter of 0.22 µm). The upper lid was then sealed, and the plate placed in a box filled with sediments from the same location for an in situ enrichment (the passage of nutrients from the original habitat to the wells occurred through the filter side of the plate). After incubation for 45 days at room temperature in the absence of light, the content of each well was re-inoculated into modified M13 plates [60] and incubated at 25 °C. A pink-colored strain, designated as ICT_E10.1, was obtained in axenic culture after 2 months and cryopreserved in glycerol 20% (*v/v*) in modified M13 medium. 

### 2.2. Partial 16S rRNA Gene Sequence-Based Phylogeny

The phylogeny of the novel isolate was first inferred with the analysis of the 16S rRNA gene sequence. Genomic DNA was obtained using the kit E.Z.N.A. Bacterial DNA Isolation Kit (Omega BIO-TEK Norcross, GA, USA) and the 16S rRNA gene sequence amplified by PCR with the universal primers 27F and 1492R [61], using a protocol previously described [26]. Purification was achieved with the illustra™ GFX™ PCR DNA and Gel Band Purification Kit (Cytiva, Marlborough, MA, USA) and the sequencing was performed at Eurofins Genomics. The Geneious software version R11 (Dotmatics, Bishop’s Stortford, UK)was used to obtain the partial 16S rRNA gene sequence, which was deposited in the National Center for Biotechnology Information (NCBI) under the GenBank accession number OL684514. The phylogeny was inferred with the use of the 16S rRNA tool of the EZBioCloud platform (https://www.ezbiocloud.net/ (accessed on 26 September 2022)) [62] and a phylogenetic tree constructed with MEGA version X (https://www.megasoftware.net/ (accessed on 26 September 2022)) [63] to show the position of strain ICT_E10.1^T^ in the family. The 16S rRNA gene sequences of the closest relatives and other representative strains in the family were taken from the NCBI database, aligned with the CLUSTALW option with MEGA [64] and the maximum likelihood phylogenetic tree constructed with 1000 bootstraps replicates, the general time reversible model and the activated gamma distributed with invariant sites (G+I) option. 

### 2.3. Genome-Based Phylogeny and Genomic Analyses 

The sequencing library was prepared using the Illumina DNA Prep library preparation kit (Illumina, San Diego, CA, USA) according to the manufacturer’s instructions and subsequently quantified using Bioanalyzer (Agilent Technologies) and QUBIT dsDNA HS (Thermofisher, Waltham, MO, USA). The final 4nM pool library was sequenced on the Illumina MiSeq instrument (Illumina, San Diego, CA, USA) using MiSeq reagent kit v.3 (Illumina, San Diego, CA, USA) for 250 cycles. The raw sequencing data trimming and filtering (min length 50 bp) were followed by genome assembly using CLC Workbench Genomics version 21.0.1 (QIAGEN, Hilden, Germany) (minimum contig length of 1000 and length/similarity fraction of 0.9). Gene calling and annotation were performed by Prodigal version 2.6.3 (http://compbio.ornl.gov/prodigal/ (accessed on 26 September 2022)) [65] and Prokka version 1.14.6 (http://vicbioinformatics.com/) [66], respectively (default parameters), and the genome quality was assessed using checkM version 1.20 (http://ecogenomics.github.io/CheckM (accessed on 26 September 2022)) [67]. The genome is deposited at NCBI (DDBJ/ENA/GenBank) with the tag JANZKV000000000.

Other genome-based markers were additionally employed according to Chun and colleagues [68] for consolidation of the phylogeny of strain ICT_E10.1^T^. The full-length 16S rRNA gene sequence was obtained from the genome using the ContEst16S tool of the EZBioCloud platform (https://www.ezbiocloud.net/) [69] and compared with other type strains as described previously. The gene coding for the beta subunit of bacterial RNA polymerase (*rpoB*) was extracted from the genome of strain ICT_E10.1^T^ and from other publicly available genomes (in NCBI) and compared as proposed previously [70,71]. The digital DNA–DNA hybridization (dDDH) values were obtained using formula d4 (GGDC formula 2) of the Genome-to-Genome Distance Calculator (GGDC) (DSMZ, Braunschweig, Germany), available at the Type Strain Genome Server (TYGS) [72,73]. The average nucleotide identity (ANI) was calculated using the OrthoANI tool in EZBiocloud (https://www.ezbiocloud.net/) [74] and the average amino-acid identity (AAI) with the AAI-profiler (http://ekhidna2.biocenter.helsinki.fi/AAI (accessed on 26 September 2022)) [75]. A genome-based tree (multi-locus species sequence-based tree—MLST) was built using the genomes of strain ICT_E10.1^T^ and of closely related strains and constructed using the autoMLST: Automated Multi-Locus Species Tree pipeline (https://automlst.ziemertlab.com/) with selected default genes and IQ-TREE Ultrafast Bootstrap analysis (with 1000 replicates) [76].

Other genome-based analyses were performed for strain ICT_E10.1^T^ and the current species with validly described names in the genus *Stieleria*. These include the genome mining for biosynthetic gene clusters (BGCs) using the AntiSMASH version 6.0 (https://antismash.secondarymetabolites.org/ (accessed on 26 September 2022)) with strict detection and all extra features enabled [77,78] and the search for putative carbohydrate-active enzymes (CAZymes) using the dbCAN2 meta server (http://cys.bios.niu.edu/dbCAN2 (accessed on 26 September 2022)) for automated carbohydrate-active enzyme annotation using all tools available [79]. 

### 2.4. Morphological and Physiological Characterization

The assays for morphological and physiological characterization were unless otherwise stated, performed at 25 °C for 14 days in modified M14 medium [60] (biomass production was visually enhanced in this medium). 

The morphological features of strain ICT_E10.1^T^ were evaluated through bright field microscopy and transmission electron microscopy (TEM) using exponentially growing cells. Bright-field images were acquired using Nikon Eclipse Ci equipment. (Nikon, Tokyo, JP)Additionally, 100 individual cells were measured with the software ImageJ (https://imagej.nih.gov/ (accessed on 26 September 2022)) to obtain the mean cell size. For TEM, colonies growing on agar M14 medium were submersed for 2 h in a glutaraldehyde (2.5% *v/v*) mixture prepared in marine buffer (pH 7.2) [26] for fixation of cells. Osmium tetroxide (1% *v/v*) diluted in the same buffer was added for an overnight (12–15 h) treatment following incubation with uranyl acetate (1% *v/v*) for 1 h. Dehydration of cells was achieved using a graded ethanol series. In the end, cells were treated with propylene oxide and embedded in epoxy resin. Ultrathin sections were obtained and stained with uranyl acetate (1% *v/v*) for 10 min followed by 10 min in Reynolds lead citrate and observed in a 100CXII transmission electron microscope (JEOL, Tokyo, Japan). Colonies of strain ICT_E10.1^T^ were also photographed for their phenotypic characterization.

The respiration mode of strain ICT_E10.1^T^ was evaluated with GENbox microaer and GENboxanaer sachets (BioMérieux, Marcy-l’Étoile, FR) using a GENbox Jar and results were visually recorded after 14 days. The assays for the evaluation of the pH range for growth, the salinity tolerance, and the carbon and nitrogen sources were performed in triplicates in a 96-well plate format using liquid cultures. For each tested condition, a pre-inoculum of strain ICT_E10.1^T^ was diluted in the proportion of 1:10 in the respective medium for a final volume of 100 µL culture per well, and results were recorded by measuring the culture turbidity (optical density (OD) at 600 nm) before and after 14 days incubation at 25 °C. The pH values tested ranged from 4.0 to 11.0 and the tolerance to NaCl was from 0 to 10 % (*w/v*) (media formulated as described previously [36]). The carbon sources assayed (0.1% *w/v*) included *N*-acetylglucosamine (NAG), cellobiose, galactose, glucose, xylose, carrageenan, mannitol, fructose, lactose, dextran and arabinose, and the nitrogen sources tested (0.1% *w/v*) were NAG, serine, alanine, cystine, tyrosine, tryptophan, phenylalanine, valine, arginine, lysine, histidine, sodium nitrite, sodium nitrate, glutamine, asparagine, urea, ammonium sulfate, and casamino acids. All these media were prepared as previously described [36]. The catalase test consisted of reacting a colony of strain ICT_E10.1^T^ with a drop of hydrogen peroxide, which lead to the formation of bubbles and indicated the presence of the enzyme. The temperature growth range was assayed in solid format: 10 µL drops of inoculum of strain ICT_E10.1^T^ were placed above modified M14 medium plates [60], incubated at 5, 10, 15, 20, 25, 30, 37, and 40 °C and results visually documented after 14 days. Following the same approach, the evaluation of vitamin requirement was tested in M14 medium in the absence of vitamins or only supplemented with the vitamin cobalamin (B_12_) (final concentration 1 mg/L) and results visually recorded after 14 days.

### 2.5. Extraction of Metabolites and Antimicrobial Screening

For extraction of metabolites, strain ICT_E10.1^T^ was cultured for 14 days, at 25 °C, under 180 rotations per minute (rpm), in 250 mL of 1:10 M13 oligotrophic medium formulated as described previously [29], slightly modified to contain sea salts^®^ (Sigma-Aldrich, St. Louis, MO, USA) (3.3% *w/v*) in replacement of the natural sea water. Afterwards, cells were collected by centrifugation (3600 rpm for 5 min in an 5810R Centrifuge (Eppendorf, Hamburg, Germany), suspended in acetone (1:1), subjected to cell disruption by sonication and the mixture collected by filtration and dried in a rotatory vacuum evaporator (Rotavapor^®^ R-100 equipment from BUCHI, Flawil, CH) to obtain a solid extract. These residues were then dissolved in 20% (*v/v*) dimethyl sulfoxide (DMSO).

The antimicrobial screening was performed against two relevant representatives of Gram-negative and Gram-positive bacteria (*Escherichia coli* ATCC 25922 and *Staphylococcus aureus* ATCC 29213, respectively), in liquid format using 96-well plates as described previously [26,80]. Briefly, the extract was diluted 1:10 with a standardized bacterial target culture (5.0 × 10^5^ colony forming units (CFU)/mL in nutrient broth medium (NB) as formulated previously [26,80]) for a final volume of 100 µL per well (crude extract concentration of 1mg/mL in the assay). Internal plate controls were added: target bacteria growth control, positive control of ampicillin (concentration of 4 mg/mL in the assay), medium control (NB) and solvent control of DMSO (concentration of 2% (*v/v*) in the assay). Culture turbidity was measured at 600 nm before and after 24 h of incubation at 37 °C and the percentage of growth calculated as described previously [26]. Three independent biological replicates were made (*n* = 3). 

## 3. Results and Discussion

### 3.1. Phylogeny of Strain ICT_E10.1^T^ and Genomic Features

The analysis of the partial 16S rRNA gene sequence showed that strain ICT_E10.1^T^ is affiliated with the genus *Stieleria* of the phylum *Planctomycetota* (Figure 1). This genus belongs to the family *Pirellulaceae* of the order *Pirellulales* and the class *Planctomycetia*. Based on the analysis of the full-length 16S rRNA gene sequence, the current closest validly described relatives of strain ICT_E10.1^T^ are *Stieleria neptunia* Enr13^T^ [52] (with 98.8% similarity), *S. maiorica* Mal15^T^ [56] (98.4%) and *S. varia* Pla52n^T^ [59] (95.8%). The threshold proposed for the delineation of novel species (98.7%) [81] is only slightly surpassed when compared with *S. neptunia* Enr13^T^. Nevertheless, the sole use of the 16S rRNA gene sequence was proved in various cases to be insufficient for inferring the phylogeny in the phylum. New planctomycete species with higher 16S rRNA gene sequence similarities (99.0–99.9%) were recently proposed based on the analysis of additional phylogenomic markers, including in the *Stieleria* genus [19,52,82,83]. All phylogenetic markers employed in this study support the affiliation of strain ICT_E10.1^T^ to the genus *Stieleria* but as a separated species, as all values obtained (Table 1) are below the threshold values proposed for new species: 71.4–88.5% for ANI and 67.8–90.6% for AAI (threshold value for both is 95% [84,85]), 23.2–37.0 for dDDH (threshold value of 70% [73]) and 82.0–94.1% for the comparison of the *rpoB* gene sequence (species threshold value is 95.5% [71]). The genome-based taxonomic analysis conducted through the TYGS also supports that strain ICT_E10.1^T^ forms a novel species within the genus *Stieleria*. The genome-based tree shows that strain ICT_E10.1^T^ clusters near *S. neptunia* Enr13^T^ but in a separate branch (Figure 2).

The main genomic features of strain ICT_E10.1^T^ are presented in Table 1 for comparison with the current species with validly described names in the genus *Stieleria*. The genome of strain ICT_E10.1^T^ obtained in this study is constituted by 455 contigs (N50 of 39,335) and according to CheckM, has a completeness of 99.93% and 0% contamination. The genome size is 9,813,311 base pairs (bp), lower than the genome sizes of *S. neptunia* Enr13^T^ (approximately 11.0 Mb, currently the biggest in the family *Pirellulaceae*) and *S. maiorica* Mal15^T^ (approximately 9.9 Mb). The DNA G+C content of strain ICT_E10.1^T^ is 58.8 in mol%, overall similar to others in the genus. According to the genome annotation with Prokka, the number of predicted protein-encoding genes and tRNAs in strain ICT_E10.1^T^ is 6964 and 109, respectively. Additionally, 4578 proteins were putatively annotated as hypothetical, which indicates that a substantial part of the genome (around 66%) is associated with proteins with still unknown functions, similar to most members in the phylum (often more than 40% of hypothetical proteins in planctomycete genomes) [8].

The additional genome-based metabolism analysis performed for strain ICT_E10.1^T^ and its closest relatives included the search for CAZymes. These are important enzymes that catalyze glycosidic bonds in carbohydrates or that display adhesion functions [86]. The putative CAZyme profile obtained for strain ICT_E10.1^T^ and its relatives is presented in Table 2 for comparison. In total, 438 CAZymes were putatively detected in the genome of strain ICT_E10.1^T^. These include diverse families of structural types of CAZymes with different functions (Table 2). The higher number of CAZymes identified was assigned to the glycoside hydrolase and glycosyltransferase families, which have the capacity of hydrolyzing or forming glycosidic bonds [86], respectively, but other families were also detected (carbohydrate esterases, auxiliary activities, carbohydrate-binding modules, polysaccharide lyases, cohesins and enzymes with undetermined functions). The profiles obtained for the other species in the genus *Stieleria* are overall similar, with the number of CAZymes proportional to the genome size (Table 2). In resume, strain ICT_E10.1^T^ and the other current species in the genus *Stieleria*, are, as many other planctomycetes, most likely capable of degrading complex sugars. 

### 3.2. Ecology

Strain ICT_E10.1^T^ was isolated from sediments collected in the “Sea of Straw” bay in the Tagus river estuary (Portugal), a typical brackish environment. Deeper ecological evidence of strain ICT_E10.1^T^ in other regions was obtained by searching for related *Planctomycetota* in the SILVA SSU database against the full-length 16S rRNA gene sequences available (release 138.1 from 27 August 2020) [41], considering hits with more than 98.7% similarity. At the present time, 56,166 sequences in the SILVA database are taxonomically assigned to the family *Pirellulaceae* and, out of these, ten metagenomes obtained from seawater showed phylogenetic proximity to strain ICT_E10.1^T^. The other type strains in the genus *Stieleria* were also all isolated from marine environments from different regions of the globe (Table 3). Furthermore, additional isolates that show proximity to the genus *Stieleria* include a strain isolated from a marine sponge collected in Moreton Bay (Australia) [87], a strain isolated from the deep sea [10], a strain isolated from brackish water samples and a strain obtained from a seawater/sediments mixture [88]. A phylogenetic tree was constructed to summarize the ecology of ICT_E10.1^T^-related *Planctomycetota* and other strains closely related to the genus *Stieleria* (Figure 3). In summary, strain ICT_E10.1^T^ and its closest uncultured relatives appear to be associated with brackish and marine ecosystems and therefore adapted to tolerate salt, as well as the other members of the genus *Stieleria* and most *Pirellulaceae* (only two genera in the family are currently associated with freshwater environments: *Pirellula* and “*Anatilimnocola”* [18,47,89,90]).

### 3.3. Physiological and Morphological Features 

The main morphological and physiological features of strain ICT_E10.1^T^ are presented in Table 3 for comparison with the current validly described species in the genus *Stieleria*. 

Physiologically, strain ICT_E10.1^T^ is heterotrophic, aerobic, catalase positive, and mesophilic. The temperature growth range is between 20 to 30 °C (optimal at 25 °C). In comparison, the other type strains in the genus have more tolerance to cold (*S. neptunia* Enr13^T^ and *S. maiorica* Mal15^T^ grow from 9/11 °C, respectively), or higher temperatures (*S. neptunia* Enr13^T^ up to 35 °C*, S. maiorica* Mal15^T^ up to 37 °C and *S. varia* Pla52n^T^ up to 45 °C) (Table 3). Strain ICT_E10.1^T^ grows from pH 6.5 to 11.0, a higher tolerance to basic pH than its closest relatives. Strain ICT_E10.1^T^ requires salt to grow and can tolerate up to 3% (*w/v*) NaCl. No vitamins are needed for growth, however, supplementation with cobalamin or a vitamin cocktail enhanced biomass production, similarly to other planctomycetes [9]. Strain ICT_E10.1^T^ is able to grow in diverse nitrogen sources (0.1% *w/v*), such as peptone, yeast extract, NAG, ammonium sulfate, casamino acids, urea, sodium nitrate, asparagine, glutamine, histidine, phenylalanine, tryptophan and alanine but not serine, cystine, tyrosine, valine, arginine, lysine, and sodium nitrite. Strain ICT_E10.1^T^ also grows using diverse carbons sources (0.1% *w/v*), such as NAG, cellobiose, galactose, fructose, lactose, arabinose, xylose, and glucose but not mannitol, dextran, and carrageenan.

Morphologically, strain ICT_E10.1^T^ colonies are highly viscous in consistency and pink pigmented (Figure 4a), as species *S. neptunia* and *S. maiorica* but not as *S. varia* (the only species in the family *Pirellulaceae* to show orange pigmentation [59]). Strain ICT_E10.1^T^ is spherical to slightly oval in shape and cells are around 1.7 µm ± 0.3 × 1.4 µm ± 0.3 in size (Figure 4b–d), overall similar to *S. neptunia* but slightly rounder and larger. Cells divide by budding and are motile in earlier stages of the life cycle (Figure 4d). In ultrathin sections, cells of strain ICT_E10.1^T^ show a planctomycete cell plan with an outer membrane, a cytoplasmatic membrane with many invaginations, ribosomes, and DNA permanently condensed (Figure 5). Crateriform pits are present in the other members of the genus *Stieleria* but in the TEM images of strain ICT_E10.1^T^, these were not observed (Figure 5). Large fibrillar structures, fimbriae, with ring-like formations at the base are present, mainly around the cell pole. These are common in other *Pirellulaceae* [14]. Cells cluster in large aggregates but also in small chains (Figure 4c), which were only observed in *S. varia*. Additionally, a strong holdfast structure was also observed connecting the cells (Figure 5a). 

### 3.4. Genome Mining for BGCs and Antimicrobial Screening

Due to the environmental relevance of the genus *Stieleria* demonstrated in previous studies [52,56], we evaluated the biotechnological potential of strain ICT_E10.1^T^ by genome mining for biosynthetic gene clusters and antimicrobial screenings against two potentially pathogenic bacterial targets. 

According to the genomic analysis with AntiSMASH v.6.0, strain ICT_E10.1^T^ genome putatively encodes for nine BGCs belonging to eight structural types: one type I polyketide synthase (PKS)/non-ribosomal peptide synthase (NRPS) hybrid, three terpenes, one acyl-amino acid, one class V lanthipeptide, one type I PKS, one NRPS/PKS-I hybrid, one type III PKS, and one NRPS. According to the analysis performed with the same AntiSMASH version on the other members of the genus *Stieleria*, the number of BGCs obtained varied between 7–9 (Figure 6). In comparison with its relatives, strain ICT_E10.1^T^ shows a different BGC profile and a higher variety of structural types (8 versus 6). The only match against any known cluster (search option known cluster blast) was only partially obtained (4% similarity) against the cluster responsible for the production of herboxidiene, which suggests that these BGCs in strain ICT_E10.1^T^ can potentially lead to the production of new biologically active molecules (NRPS, PKS, acyl-amino acids and lanthipeptide clusters) whereas terpene-like clusters can possibly be associated with the production of the carotenoids responsible for its pigmentation [91].

The biotechnological potential of strain ICT_E10.1^T^ was also demonstrated by the antimicrobial screenings performed. Antimicrobial activity of this strain was first evidenced in a previous study, but inhibition was mild against *S. aureus* [26]. In this study, motivated by the activity first observed and in an attempt to enhance it, we followed the one strain many compounds (OSMAC) principle [92,93] and tested a stress induction condition through oligotrophy during fermentation (1:10 M13 medium [29] versus the previously used modified M14 medium supplemented with NAG [26]). Using a similar extraction protocol with acetone as organic solvent, the ICT_E10.1^T^ crude extract obtained in this study showed, at the tested concentration (1 mg/mL), potent activity (total inhibition) against the Gram-positive bacteria *S. aureus* but not against the Gram-negative *E. coli* (Figure 7), which suggests some specificity. Strain ICT_E10.1^T^ genome also encodes for an acyl amino acid cluster, as referred previously, which is the BGC structural type putatively attributed to the production of stieleriacines in *S. maiorica* Mal15^T^ and *S. neptunia* Enr13^T^ [52,56]. However, the acyl-amino acid cluster found in strain ICT_E10.1^T^ only partially matched (37% of genes show similarity) with one of the acyl-amino acids clusters in *S. neptunia* enr13^T^ (cluster blast search analysis in AntiSMASH 6.0), which suggests that it most likely codes for the production of a putative different stieleriacine or even a chemically distinct molecule. Based on the potent antimicrobial activity demonstrated and in the genomic differences in comparison with its closest relatives, strain ICT_E10.1^T^ appears to be a promising strain regarding the putative presence of novel bioactive molecules.

## 4. Conclusions

In this study, we contributed to the knowledge of the recently created genus *Stieleria* of the bacterial phylum *Planctomycetota* by formally describing a novel species through a polyphasic approach. Genomic, morphological, and physiological comparison of our isolate strain ICT_E10.1^T^ (previously obtained from sediments collected in the Portuguese estuary of the Tagus river) with its closest relatives strongly suggests that it belongs to a novel species in the genus *Stieleria* from the family *Pirellulaceae* (order *Pirellulales* and class *Planctomycetia*), for which we propose the name *Stieleria sedimenti*, represented by strain ICT_E10.1^T^ (=CECT 30514^T^ = DSM 113784^T^) as the type of strain. Due to the biotechnological relevance of the genus *Stieleria* (from which the first and currently only planctomycetal secondary metabolites with antimicrobial activity were isolated, the stieleriacines), we explored the biotechnological potential of our strain by genome mining for biosynthetic gene clusters and by antimicrobial screenings. Potent activity against the Gram-positive bacteria *Staphylococcus aureus* was observed and diverse relevant BGCs was detected, including one belonging to the same structural type putatively assigned to the production of stieleriacines (acyl-amino acid cluster). However, the similarity between the two falls on less than 40%, which suggests that it may be able to encode for a new type of stieleriacine (or even other chemically distinct compounds). Taken together, these results show that strain ICT_E10.1^T^ is a promising strain regarding future upscaling studies aimed at the identification and isolation of bioactive compounds, including possibly novel stieleriacines.

## 5. Description of *Stieleria sedimenti* sp. nov.

*Stieleria sedimenti* (se.di.men’ti. L. gen. n. *sedimenti*, of sediment). Colonies are pink-colored and highly viscous in consistency. Cells are spherical to slightly ovoid in shape, motile, and around 1.7 µm ± 0.3 × 1.4 µm ± 0.3 in size. Cells cluster strongly in aggregates or small chains. Cell division occurs through budding and younger cells are motile. Large fimbriae with ring-like formations at the base were observed, mostly located on the cell pole. The species is heterotrophic and aerobic. Growth occurs from 20 to 30 °C, 0.5 to 3% (*w/v*) NaCl, and from pH 6.5 to 11.0. The species does not require vitamins for growth, but biomass production is enhanced. The species is catalase positive. Peptone, yeast extract, NAG, ammonium sulfate, casamino acids, urea, sodium nitrate, asparagine, glutamine, histidine, phenylalanine, tryptophan, and alanine are utilized as nitrogen sources (0.1% *w/v*) but not serine, cystine, tyrosine, valine, arginine, lysine, and sodium nitrite. NAG, cellobiose, galactose, fructose, lactose, arabinose, xylose, and glucose are consumed as carbon sources (0.1% *w/v*) but not mannitol, dextran, and carrageenan. 

The type of strain, ICT_E10.1^T^ (=CECT 30514^T^ = DSM 113784^T^) was isolated from brackish sediments collected in the Tagus river estuary (Portugal) in May 2021. The genome size is 9,813,311 bp and DNA G+C content is 58.8 mol%.

## Figures and Tables

**Figure 1 microorganisms-10-02151-f001:**
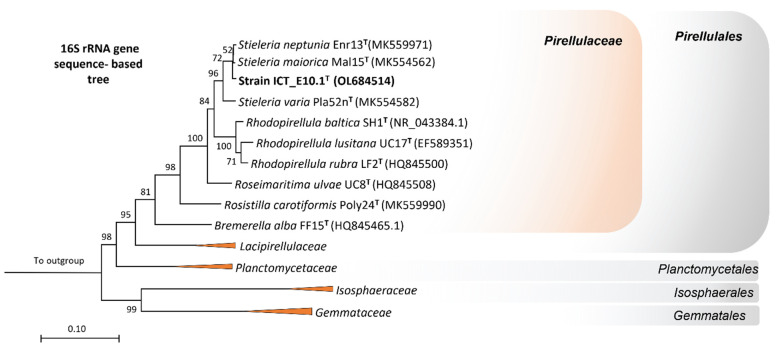
Phylogeny of strain ICT_E10.1^T^ through the analysis of the 16S rRNA gene sequence. The maximum-likelihood tree evidences the proximity of strain ICT_E10.1^T^ to the genus *Stieleria* within the family *Pirellulaceae* of the bacterial phylum *Planctomycetota*. GenBank sequence identifiers of other type strains are shown in parentheses. One thousand bootstraps were applied, and the respective values given at the nodes (in %). Branches of the other families in the class *Planctomycetia* collapsed at the family level. Three strains from the phylum *Verrucomicrobiota* were utilized as outgroups.

**Figure 2 microorganisms-10-02151-f002:**
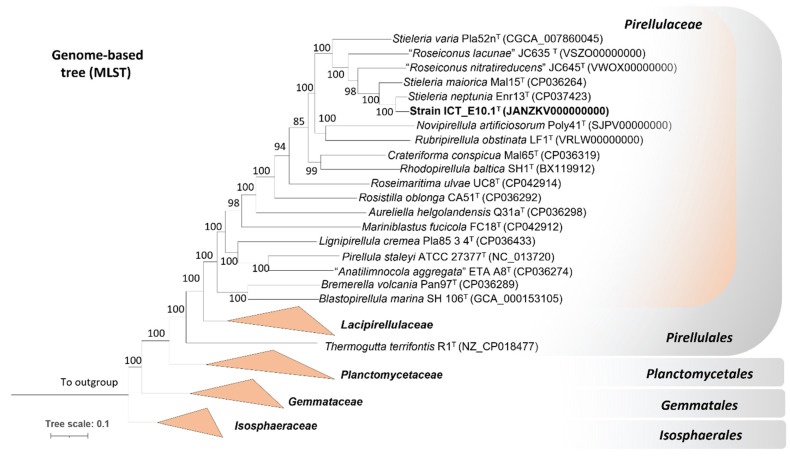
Genome-based tree (MLS) consolidating the affiliation of strain ICT_E10.1^T^ to the family *Pirellulaceae* in the *genus Stieleria* but as a separate species. Other genomes utilized in this tree were retrieved from NCBI and the GenBank tags given in parenthesis. Members belonging to other families in class *Planctomycetia* collapsed at the family level. The bootstrap used in this tree was 1000 and the respective percentages shown at the nodes (in %). Three *Streptomyces* spp. (phylum *Actinomycetota*) were used as outgroups.

**Figure 3 microorganisms-10-02151-f003:**
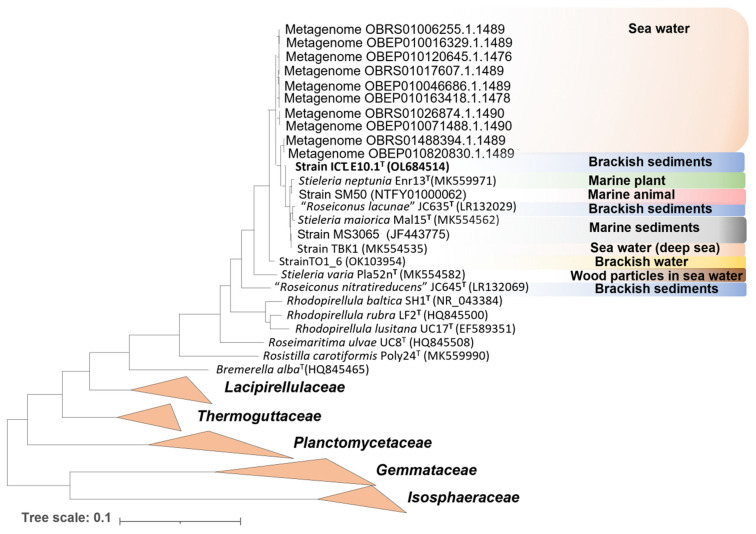
Ecological summary of ICT_E10.1^T^-related metagenomes and other isolated strains with proximity to the genus *Stieleria*. The 16S rRNA gene sequence-based tree was constructed with MEGA X as described previously. Metagenomic 16S rRNA gene sequences were retrieved from the SILVA SSU database (release 138.1 from 27 August 2020) and sequences from isolated strains retrieved from the NCBI database (GenBank accession numbers are given in parentheses).

**Figure 4 microorganisms-10-02151-f004:**
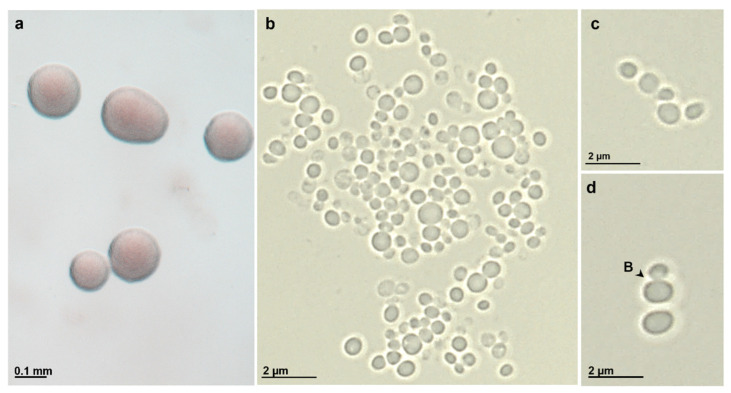
Phenotypic characterization of strain ICT_E10.1^T^ while in exponential phase. Photographies of strain ICT_E10.1^T^ in modified M14 medium (**a**) show its pink coloration and in bright field microscopy images (**b**–**d**) cells appear spherical to ovoid in shape and cluster in aggregates (**b**) or small chains (**c**). Budding is the division mode (**d**).

**Figure 5 microorganisms-10-02151-f005:**
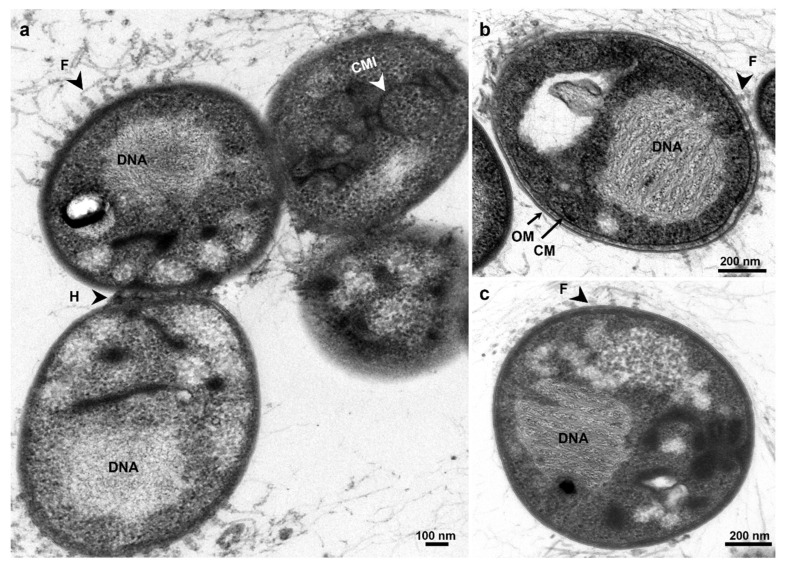
Micrographs from transmission electron microscopy observations showing the cell plan of strain ICT_E10.1^T^ in aggregated (**a**) and individual cells (**b**,**c**). A strong extracellular material is seen connecting the cells (**a**). F: fimbriae, H: holdfast, CMI: cytoplasmatic membrane invaginations, CM: cytoplasmatic membrane, OM: outer membrane.

**Figure 6 microorganisms-10-02151-f006:**
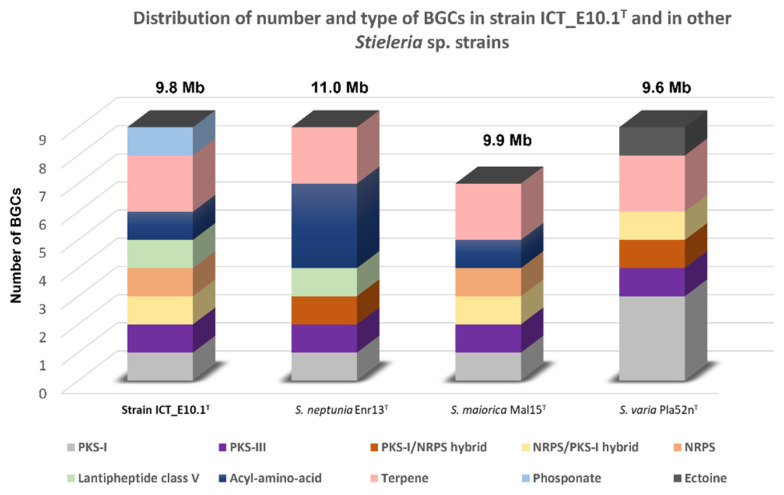
Distribution of number and structural types of BGCs putatively found in strain ICT_E10.1^T^ and in its closest relatives *S. maiorica* Mal15^T^, *S. neptunia* Enr13^T^ and *S. varia* Pla52n^T^ for comparison, evidencing the content differences between strains. The genome size of each strain was additionally added above each bar.

**Figure 7 microorganisms-10-02151-f007:**
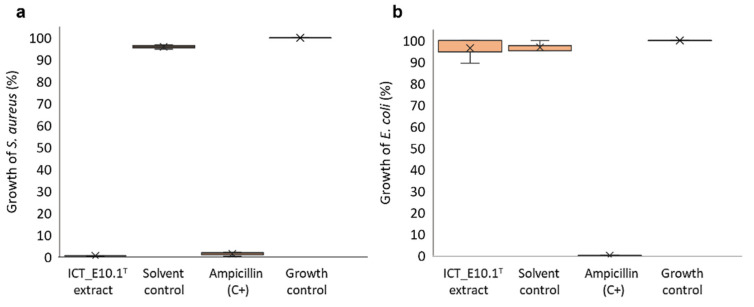
Boxplot diagram showing the percentage of growth of *S. aureus* (**a**) and *E. coli* (**b**) when exposed to ICT_E10.1^T^ extract, considering the three biologically replicated assays. The solvent control corresponds to DMSO (2% *v/v* final concentration in the assay), the positive control of ampicillin (4 mg/mL final concentration in the assay), and the growth control of the target bacteria without exposure to treatments. The planctomycete extract consistently inhibited the growth of *S. aureus* in the three assays (**a**). In contrast, very low bioactivity was observed against *E. coli* (**b**).

**Table 1 microorganisms-10-02151-t001:** Main genomic features of strain ICT_E10.1^T^ and data on the current validly described members of the genus *Stieleria* for comparison. Genomes of *Stieleria neptunia* Enr13^T^, *S. maiorica* Mal15^T^, and *S. varia* Pla52n^T^ were re-annotated in this study with the same version of Prokka used for the genome of strain ICT_E10.1^T^.

Characteristics	ICT_E10.1^T^	^a^ Enr13^T^	^b^ Mal15^T^	^c^ Pla52n^T^
Similarity (%) of the complete 16S rRNA gene sequence	na	98.8	98.4	95.8
Genome size (Mb)	9.8	11.0 *	9.9 *	9.6 *
G+C content (mol%)	58.8	58.9 *	59.3 *	56.0 *
Number of protein-encoding genes	6964	7799	6936	7000
Number of hypothetical proteins	4578	5220	4522	4848
dDDH estimated with strain ICT_E10.1^T^ (%)	na	37.0	24.0	23.2
ANI value with strain ICT_E10.1^T^ (%)	na	88.5	79.9	71.4
Similarity (%) of the *rpoB* gene with strain ICT_E10.1^T^	na	94.1	88.9	82.0
AAI value with strain ICT_E10.1^T^ (%)	na	90.6	83.1	67.8

^a^ GenBank CP037423, ^b^ GenBank CP036264, ^c^ GenBank GCA_007860045. * Data retrieved from species descriptions studies [52,56,59]. na—not applicable.

**Table 2 microorganisms-10-02151-t002:** Profile of number and families of carbohydrate-degrading enzymes putatively found in strain ICT_E10.1^T^ genome and on its closest relatives (*Stieleria neptunia* Enr13^T^, *S. maiorica* Mal15^T^, and *S. varia* Pla52n^T^) for comparison.

Family of CAZymes	Number of Enzymes			
	ICT_E10.1^T^	^a^ Enr13^T^	^b^ Mal15^T^	^c^ Pla52n^T^
Glycoside hydrolases	119	155	146	96
Carbohydrate esterases	34	45	43	46
Glycosyltransferases	118	124	130	113
Auxiliary activities	7	8	6	7
Carbohydrate-binding modules	100	94	84	79
Polysaccharide lyases	14	16	10	6
Unknown	15	13	15	14
Cohesins	1	1	1	0
Total	438	456	435	361

^a^ GenBank CP037423, ^b^ GenBank CP036264, ^c^ GenBank GCA_007860045.

**Table 3 microorganisms-10-02151-t003:** Morphological, ecological, and physiological traits of strain ICT_E10.1^T^ compared to the current validly described members of the genus *Stieleria*. Data from *Stieleria neptunia* Enr13^T^, *S. maiorica* Mal15^T^, and *S. varia* Pla52n^T^ was retrieved from species descriptions studies [52,56,59].

Characteristics	ICT_E10.1^T^	Enr13^T^	Mal15^T^	Pla52n^T^
Cell shape	Spherical to ovoid	Spherical to ovoid	Spherical to pear-shaped	Ovoid to grain rice-shaped
Cell size (µm)	1.7 ± 0.3 × 1.4 ± 0.3	1.6 ± 0.1 × 1.1 ± 0.1	1.9 ± 0.2 × 1.4 ± 0.2	1.8 ± 0.3 × 0.9 ± 0.2
Main form of aggregation between cells	Aggregates and short chains	Aggregates	Rosettes	Rosettes and short chains
Reproduction	Budding	Budding	Budding	Budding
Motility	Yes	Yes	Yes	Yes
Crateriform structures	No	Yes	Yes	Yes
Colony color	Pink	Pink	Pink	Light orange
Isolation source (location)	Brackish sediments (Portugal)	*Posidonia* sp. (Italy)	Sediments (Spain)	Wood particles in sea water (Germany)
Temperature range (°C)	20–30	9–35	11–37	20–45
pH range	6.5–11	6.5–9.0	5.5–9.0	6.0–8.0
% (*w/v*) NaCl tolerance	0.5–3	NDA	NDA	NDA
Vitamin B_12_ requirement	No	NDA	NDA	NDA
Carbon sources	NAG, cellobiose, galactose, fructose, lactose, arabinose, xylose and glucose	NDA	NAG, arabinose, cellobiose, fucose, fructose, galactose, gentiobiose, glucose, gluconic acid, glucuronamide, glucuronic acid, lactose, lactulose, mannose, melibiose, glucoside, draffinose, rhamnose, sucrose, trehalose, turanose, psicose	NDA
Nitrogen sources	NAG, peptone, yeast extract, ammonium sulfate, casamino acids, urea, sodium nitrate, asparagine, glutamine, histidine, phenylalanine, tryptophan, and alanine	NDA	NDA	NDA
Respiration	Aerobic	Aerobic	Aerobic	Aerobic

NDA: no data available.

## Data Availability

This Whole Genome Shotgun project has been deposited at DDBJ/ENA/GenBank under the accession JANZKV000000000. The version described in this paper is version JANZKV010000000. 16S rRNA gene accession number: GenBank = OL684514.

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
