# Peer review of "Stieleria sedimenti sp. nov., a Novel Member of the Family Pirellulaceae with Antimicrobial Activity Isolated in Portugal from Brackish Sediments"

_microorganisms, 2022, doi:10.3390/microorganisms10112151_

Round 1

Reviewer 1 Report

Manuscript Title: Stieleria sedimenti sp. nov., a novel member of the family

Pirellulaceae with antimicrobial activity isolated in Portugal from brackish sediments

Manuscript ID: microorganisms-1965970

The manuscript written by Vitorino et al. reports some interesting results. In this work, the authors described a novel Stieleria isolate designated as strain ICT_E10.1T, obtained from sediments collected in the Tagus estuary (Portugal). Furthermore, this isolate showed biotechnological potential by displaying relevant biosynthetic gene clusters and potent activity against Staphylococcus aureus. In general, this paper is clearly laid out, well planed and easy to read.

Some specific suggestions or questions are listed below:

1. Line 36: PVC, please use full name for the first time. Please check throughout the manuscript that abbreviations/acronyms are defined the first time they appear in each of three sections: the abstract; the main text; the first figure or table.

2. Introduction: This section needs to be reorganized. The Introduction section should briefly place the study in a broad context and highlight why it is important. It should define the purpose of the work and its significance. The current state of the research field should be reviewed, and key publications cited. Finally, briefly mention the main aim of the work and highlight the principal conclusions. However, the novelty and significance of the manuscript were not highlighted in the Introduction section, please modify the introduction more clearly.

3. Line 102: strain ICT_E10.1T, has the described strain been deposited in a public strain collection? Please mention the collection number in the manuscript.

4. Figure 6 can be improved.

5. Conclusions: This section can be revised for the better understanding of the topic and its future research.

Author Response

We are grateful for the comments provided, which helped improve the manuscript. Please see attachment for our point-by-point responses. 

Reviewer 2 Report

Article represents the description of novel planctomycete Stieleria sedimenti sp. nov, isolated from sediments in the Tagus river estuary (Portugal).

In general, the manuscript is well-written and the results meet the criteria to propose strain ICT_E10.1T as representative of a novel species Stieleria sedimenti sp.nov. belonging to the genus Stieleria. Required physiological tests, as well as genomic and phylogenomic analyses were performed for taxonomic description. In addition, antimicrobial activity of novel planctomycete was also tested against E. coli and S. aureus with the positive inhibitory effect for the latter one. 

Please, check some typo mistakes

line 52 - validly published

line 320- to tolerate salt, as well as or together with the other members

line 406 - activity of (not by) this strain

The manuscript is generally ready for publication

Author Response

(The authors gave the same response as above.)

Round 2

Reviewer 1 Report

The manuscript was improved as per the reviewer comments.